# The Response of the Replication Apparatus to Leading Template Strand Blocks

**DOI:** 10.3390/cells12222607

**Published:** 2023-11-11

**Authors:** Marina A. Bellani, Althaf Shaik, Ishani Majumdar, Chen Ling, Michael M. Seidman

**Affiliations:** Laboratory of Molecular Biology and Immunology, National Institute on Aging, National Institutes of Health, Baltimore, MD 21224, USA; marina.bellanidecairatti@nih.gov (M.A.B.);

**Keywords:** DNA replication, replisome, CMG structure, DNA replication stress

## Abstract

Duplication of the genome requires the replication apparatus to overcome a variety of impediments, including covalent DNA adducts, the most challenging of which is on the leading template strand. Replisomes consist of two functional units, a helicase to unwind DNA and polymerases to synthesize it. The helicase is a multi-protein complex that encircles the leading template strand and makes the first contact with a leading strand adduct. The size of the channel in the helicase would appear to preclude transit by large adducts such as DNA: protein complexes (DPC). Here we discuss some of the extensively studied pathways that support replication restart after replisome encounters with leading template strand adducts. We also call attention to recent work that highlights the tolerance of the helicase for adducts ostensibly too large to pass through the central channel.

## 1. DNA Replication Stress

DNA replication is driven by a multiprotein complex known as the replisome. Although they contain many proteins, replisomes consist of two fundamental functional components. These are a helicase to unwind DNA and polymerases to synthesize it. While textbooks imply a smooth and unhindered process, in reality, duplication of the genome must surmount numerous challenges. These block or slow the replication fork, inducing what is termed replication stress. Failure to overcome these impediments can result in mutagenesis and rearrangement of genomic sequences, activation of inflammatory pathways, cell senescence, or cell death. There are many sources of replication stress including non-B form DNA conformations, nucleotide insufficiency, DNA: protein crosslinks, transcription complexes, and DNA: RNA hybrids. Among the most extensively studied inducers, and with the longest history, are chemical modifications of DNA. These are formed by exposure to ultraviolet light or a host of endogenous or environmental DNA reactive compounds. If they escape one or another of the multiple repair pathways designed to remove them, they can interfere with the progress of the replisome. Here we will discuss some of the responses of the replication machinery to encounters with covalent DNA adducts.

### Replisome Structure

The structure of the replisome has been the subject of intense investigation in recent years, driven by extraordinary biochemical reconstitutions of active replisomes and the remarkable developments in cryo-electron microscopy. The impressive progress in the field has been the subject of numerous recent reviews [1,2,3,4]. 

In eukaryotic cells, the helicase has a core complex of six related, but non-identical, mini chromosome maintenance proteins (MCM 2-7) [5]. They have distinct N (N-tier) and C (C-tier) terminal regions, with the former having zinc finger and oligonucleotide/oligosaccharide-binding (OB) domains that interact with DNA, while the latter have the ATPase activities that drive DNA unwinding. The MCM subunits form a heterohexameric offset “lock washer” ring structure with a gap between MCM 2 and 5. This forms a gate through which double-strand DNA can enter [6,7,8,9]. Cell-cycle-regulated kinases drive the loading of MCM hexamers in late Mitosis and early G_1_ by the proteins of the origin recognition complex (ORC) together with Cdc6 and Cdt1 [10]. Two MCM complexes are loaded at each site such that the hexamers encircle duplex DNA in a head-to-head orientation (N-tier: N-tier), forming the double hexamer, a process called licensing [2,3,11]. 

In metazoan cells, the loading sites, and thus potential initiation sites, are typically depleted of nucleosomes and are correlated with the transcription start sites and CpG islands [11,12]. Sequences with G-quadruplex forming potential are frequently located a few hundred base pairs upstream of replication initiation sites and may reflect the tendency of ORC to be associated with nucleosome-free regions [13,14]. The greater frequency of licensing in open chromatin underlies a temporal program of replication, with euchromatin (compartment A) replicating early in the S phase while replication in heterochromatin (compartment B) is biased to the late S phase [15,16]. Early replicating regions tend to be in the nuclear interior, while late replication occurs near the nuclear lamina and nucleoli [17].

The licensing process is tightly regulated such that it cannot occur in the S phase, the result of destruction and/or inhibition of licensing factors by cyclin-dependent kinases that are active in the S phase [18]. This eliminates the possibility of re-replication, thus preserving gene copy number, but at the potential expense of a fallback option for completing replication in the event of stalled or blocked forks. However, this limitation has been overcome by licensing 10–20 times more origins than are used in a relatively stress-free S phase. These form a reserve of dormant origins that can be activated in response to replication stress [19]. Multiple licensed origins are distributed in domains extending over 400–800 kilobases, although the site of loading may not correspond to an eventual origin of replication since the complex can be moved by the transcriptional apparatus to distal locations [20,21]. The stochastic firing of origins within domains has been the long-held view [22,23], but conflicting conclusions have been advanced in recent experiments that demonstrate control of initiation by cis-acting sequences [11,24]. It may be that some feature of the G/C rich character of origin regions plays an important role. 

Recent cryo-electron microscopy of the human MCM complex isolated from HeLa cells shows a central channel encircling DNA that extends through the entire length of the two hexamers. The principal interactions of the MCM subunits are with the leading template strand [25]. A single base pair is disrupted at the interface of the two hexamers [25]. In this form, the complexes are stable but nonfunctional. Activation of the hexamers requires a series of reactions mediated by specialist kinases DDK (Dbf4-dependent kinase) and CDK (cyclin-dependent kinase) [26]. Phosphorylation of MCM4 and MCM6 by DDK enables binding by TRESLIN/TICRR (TopBP1-interacting, replication-stimulating protein, Sld3 in yeast) in complex with MTBP (MDM2 binding protein, Sld7 in yeast) and the recruitment of CDC45. Phosphorylation by CDK of additional intermediates results in the association of Pol ε (E) and the GINS complex, a group of four small proteins, SLD5, PSF1, PSF2, and PSF3 (Go, Ichi, Ni, and San: Japanese for five, one, two, and three) [27]. As shown in recent work from several laboratories, DONSON, a dimeric non-enzymatic scaffold protein, is required for loading GINS and Pol ε onto each MCM2-7 complex [28,29,30,31,32,33]. This also stabilizes the interaction with CDC45, resulting in the formation of the CMGE (**C**DC45-**M**CM-**G**INS-**P**OLε) in which the MCM2-5 gate is closed and locked around duplex DNA by CDC45 and GINS [34,35,36]. The junction between the MCM hexamers is disrupted and they are rotated away from each other such that the central channels are no longer aligned [37,38].

Cryo-electron microscopy analyses of the CMGE reconstituted from recombinant yeast proteins and loaded on duplex DNA show that association with Pol ε, acting as a structural factor [39], causes the formation of a short bubble of melted DNA in each CMGE [38,40]. The discrepancy regarding the location of disrupted base pair(s) in the human double hexamer (at the junction) and the yeast CMGE (within each CMGE) is unresolved at this time although it could be due to a species difference, or the recombinant vs. endogenous source of the proteins, or a difference between the MCM double hexamer and the more complex CMGE [25]. The two fully active helicases, each encircling what will become the template strand for leading strand synthesis, and leading with the N terminal tier, move 3′-5′ towards, and then past one another on opposite strands, thus driving bidirectional replication [41].

The mechanism of DNA unwinding has been of considerable interest. An additional protein, MCM10, associates with the CMGE and stimulates DNA unwinding [42,43,44,45]. Double-strand DNA is drawn into the N-terminus of the N-tier where it is ringed by zinc fingers in MCM6, 4, and 7. The duplex DNA is pulled through the central channel until motifs from MCM 7, 4, and 6 at the base of the N-tier block further progress the lagging template strand [2,46,47]. This “steric exclusion” of the lagging template strand forces it from the central channel through an opening between MCM3 and MCM5 [46,47,48,49,50,51] onto the surface of the MCM complex where it is guided towards the polymerase α-primase [52]. Primers are synthesized on the lagging strand of each replisome, then extended by polymerase δ (the lagging strand polymerase), after which there is a polymerase switch and polymerase ε, located at the C’ tier of the CMG at the exit of the leading template strand, takes on leading strand synthesis [1,3,52]. The localization of the polymerase α-primase to the exit channel of the lagging strand template explains the efficient priming of lagging strand synthesis and the very inefficient priming of leading strand synthesis by this enzyme. 

Certain features of this model of origin activation and lagging strand eviction have drawn comment. In a recent discussion, based on work with the SV40 T antigen helicase, Li, O’Donnell, and colleagues note that the head-to-head orientation of each CMG means that as long as they encircle duplex DNA, they cannot move past each other [53]. Furthermore, the size of each CMG would require the unwinding of about four turns of duplex DNA, substantially greater than the few base pair bubbles observed in the recent cryo-em study [38]. Instead, they suggest that each CMG pulls the leading template strand towards itself. Since the two are pulling on opposite strands, the interactions between the strands will be disrupted (they refer to this as “shearing”), with the resulting single strands passing out through the C terminal exit channel of each CMG. Eventually, enough single-strand DNA accumulates to allow occupancy by individual CMGs, subject to the ejection of the lagging template strand, which, of course, is the leading template strand of the other CMG. This event sequence builds the bidirectional replication bubble which then expands by translocation of the helicases. 

Langston et al. also address the ejection of the lagging template strand. While various groups have proposed the existence of a formal gate between individual MCM proteins (see above), they reference earlier work demonstrating the loading of hexameric helicases, including the CMG, on single-strand DNA without accessible ends [49,54,55]. They argue that breathing of the interaction surfaces of the helicase subunits would allow passage of single-strand DNA into or out of the central channel of the CMG, obviating the need for a gate between specific subunits [53]. 

Upon the formation of the origin bubble, the two replisomes move away from one another. The mechanism of translocation by the hexameric replicative helicases has been deduced with monomeric complexes from T7 bacteriophage, which moves along the lagging strand template in the 5′-3′ direction [50], the archaeal Saccharolobus solfataricus which progresses along the leading strand template in the 3′-5′ direction [51], and the Drosophila CMG [48]. Translocation is by a “hand over hand” mechanism in which ATP hydrolysis drives the sequential movement of each subunit from one end of the complex to the other end of the offset ring (also described as a spiral staircase). Interactions between individual subunits, and between subunits and DNA, form and are broken and then reformed as the subunits move from the base to the top of the spiral staircase. Thus, although the MCM 2-7 ring is “locked” around the leading strand template by CDC45 and GINS, the individual subunits are not locked to each other, a feature that is likely to offer some flexibility when blocks are encountered. 

A fully functional replisome capable of leading strand synthesis in biochemical experiments can be assembled with 43 peptides [56]. However, the situation in living cells is far more complicated (e.g., BioGrid lists more than a thousand partners of MCM2). Associated factors stimulate DNA synthesis, resolve impediments to the CMG such as alternate DNA structures and protein: DNA complexes, activate replication stress response pathways, modify histones, and repair and restore stalled and broken forks, etc. [57,58]. Among numerous additional factors are RPA and the members of the fork protection complex (FPC), including TIMELESS-TIPIN, AND-1, and CLASPIN. TIMELESS-TIPIN binds the N terminal tier of the MCM complex. They make first contact with the parental double-strand DNA and guide it into the helicase [4,47,59]. TIMELESS-TIPIN stabilizes the binding of CLASPIN, which interacts with MCM2 and MCM6 at the front, and with Pol ε at the back of the replisome, thus linking unwinding and leading strand synthesis. AND-1 contacts both CDC45 and GINS and stimulates DNA synthesis through its DNA binding domain [47,56]. SDE2 stabilizes TIMELESS and the FPC and protects reversed forks from degradation [60]. The histone chaperone FACT (facilitates chromatin transcription) associates with the CMG and Pol α and is required for the replication of chromatin, perhaps by disrupting chromatin ahead of the replisome [61]. RECQL4, a binding partner of MCM10 and AND-1, is involved in the formation of the CMG and initiation of replication [62,63]. Accessory helicases also interact with replisome components. These translocate in the 5′-3′ direction, thus on the lagging template strand, and resolve various impediments to the replisome. For example, RTEL1 unwinds R-loops and G-4 structures, disrupts non-covalent nucleoprotein complexes, and is involved in telomere maintenance [64,65].

## 2. What Happens When the Replisome Encounters DNA Damage? 

Covalent DNA adducts can be on either the leading or lagging template strands or if interstrand crosslinks (ICLs), on both. The current understanding of replisome structure implies that strands with relatively small adducts can be drawn into the MCM2-7 entry channel and either be evicted if on the lagging template strand or continue through if on the leading template strand. As inhibitors of the relevant polymerases, they will be left behind in gaps in the daughter strands to be dealt with by translesion polymerases and appropriate repair pathways. However, if large base adducts, such as DNA: protein crosslinks (DPC) [66], enter the channel they would be expected to stall the helicase. Here we will discuss a reconsideration of this expectation as well as three well-characterized pathways that follow the uncoupling of the helicase and the leading strand polymerase. 

## 3. Polymerase Blocks

### 3.1. Fork Reversal

Fork reversal is another option when replisomes encounter blocks (Figure 1A). Reversed forks are described as a chicken foot, a four-stranded structure formed by a combination of hybridization of daughter strands to each other and the rehybridization of the parental template strands. Models of reversed forks often show a leading strand lesion embedded in the rehybridized parental duplex as an intermediate in pathways that support the completion of replication. Indeed, the teleological argument for reversed forks is that they enable adduct repair, template switching, or the completion of synthesis by the arrival of a fork coming from a downstream origin [67].

Reversed forks are induced by multiple genotoxic agents, including topoisomerase inhibitors, base adducts, ICLs, and inhibitors of DNA synthesis such as hydroxyurea (HU) [68]. Formation requires contributions from multiple factors including DNA translocases, nucleases, helicases, recombinational activities, and topoisomerases [69,70,71]. There are two scenarios in which they occur. In one the helicase is blocked by lesions that cannot pass through the channels of the MCM complex. In the other, the helicase is uncoupled from the activity of Pol ε either by a lesion on the leading template strand that is channel compatible but polymerase inhibitory or by the unavailability of dNTPs. The two situations differ in the sequence of events that precede the actual fork reversal, but an important question for both has been the disposition of the replisome during the process.

Fork reversal induced by a helicase-blocking lesion has been examined in a study from the Walter laboratory. Amunugama and coworkers followed the replication of plasmids containing a defined cisplatin ICL in Xenopus egg extracts. In this system, two forks moving in opposite directions eventually collide at the ICL. Fork convergence is required for removal of the replisomes after which repair can proceed [72]. Remarkably, after a 45 min incubation, almost 60% of the forks were reversed, a result that was dependent on the unloading of the replisomes [73]. The authors proposed that the eviction of the replisome permitted loading of the translocases such as SMARCAL1, ZRANB3, and HLTF on the revealed single-strand DNA and access to the 3′ hydroxyl of the leading daughter strand. However, they also noted the reversal of single forks in other experimental systems, indicating that convergent forks are not required [74,75]. In the single fork experiments, cells were exposed to agents, such as HU, that induced uncoupling of the CMG helicase and DNA synthesis. This would generate stretches of single-strand DNA between the stalled leading daughter strand and the still progressing CMG helicase. They suggested that the single-strand DNA would allow the assembly of the reversal factors without necessarily requiring the removal of the replisome. In the Xenopus egg extract system, equivalent stretches of single-strand DNA are not formed in the crosslinked plasmid since the replisomes are stalled at the block and occupy some 24 nucleotides of the leading strand template [72]. Thus, the requirement for unloading the replisome. 

Recently, the Cortez laboratory addressed the question of replisome retention or loss on forks reversed by HU treatment [76]. Although HU does not generate an adduct requiring repair, and there can be no convergent synthesis to complete replication, it is quite effective at inducing the conditions that favor fork reversal—stretches of single-strand DNA behind the CMG, leaving an accessible 3′ hydroxyl on the stalled leading daughter strand. Fork reversal was dependent on the activity of the SMARCAL1 translocase and the strand exchange activity of RAD51. Furthermore, the replisome was retained on the DNA, consistent with the requirement for the CMG to restart DNA synthesis upon removal of HU. They proposed a clever model in which RAD51 would generate a paranemic hybrid between the two parental single strands. (The two strands interact with each other but are not interwound. Unlike the conventional DNA double helix, strand separation would not require rotation of one strand around the other [77].) This would trap the replisome on the template strand for leading strand synthesis in a bubble between the parental paranemic hybrid and the parental duplex DNA ahead of the fork. The translocase would drive branch migration and hybridization of the two daughter strands to form a reversed fork. This would leave the replisome at the original fork available to continue synthesis when conditions permit. The suggestion of a paranemic interaction between the unwound parental strands is intriguing. The formation of paranemic hybrids by recombinases has been known for decades [78,79,80]. While they have received recent attention for applications in DNA-based nanotechnology [81,82,83], the lack of facile methods for their identification in cellular systems has limited exploration of their physiological relevance. Perhaps the report from the Cortez group will stimulate interest in developing detection reagents suitable for biological experiments.

Although the frequency of reversed forks in these two experimental systems was high, levels vary in different experimental conditions, likely depending on the balance of the competition with other pathway options [84]. For example, in an analysis by electron microscopy, reversed forks in UV-treated cells accounted for more than 20% of all forks [68], while in other studies, also based on electron microscopy, the frequency was less than 4% and not influenced by UV treatment [85,86]. 

### 3.2. Lesion Skipping and Repriming

The classic example of leading strand adducts that can pass through the single strand channel of the CMG but are unreadable by replicative polymerases are ultraviolet cyclopyrimidine (CPD) photoproducts [87,88]. The effect of UV on replication was examined decades ago in a foundational experiment by Rupp and Howard-Flanders. They found in excision repair-deficient *E. coli* cells that newly synthesized single-strand DNA was in long tracts, while exposure of cells to UV resulted in the appearance of the newly replicated DNA in smaller fragments. Over time the smaller fragments became much longer. They interpreted their results as evidence for blockage of synthesis by UV photoproducts followed by a restart of synthesis that continued until another encounter with a photoproduct, thus forming the short fragments (Figure 1B). Eventually, recombinational repair processes filled gaps and joined the fragments to complete replication [89]. Later, Marians and coworkers demonstrated skipping of leading strand UV damage and repriming by the *E. coli* replication complex [90]. Similar observations in mammalian cells were made by Lehmann [91]. In a direct demonstration of the uncoupling of the helicase and the polymerase in vivo, electron micrographs of DNA isolated from yeast and mammalian cells exposed to UV showed replication fork-associated gaps in both leading and lagging daughter strands [85,92]. In biochemical experiments, a reconstituted yeast replisome could translocate past a CPD. However, consistent with its location on the CMG, repriming by the lagging strand Pol α primase/polymerase was inefficient [52,93]. Eukaryotic cells overcome this limitation by expressing another primase/polymerase, PrimPol, which has translesion polymerase activity against UV photoproducts but also can reprime synthesis on single-strand DNA generated by the continued progression of the CMG uncoupled from Pol ε [94,95,96]. Many activities involved in replication and the response to replication stress are tightly regulated. It has been shown that BRCA2, in association with MCM10, controls PrimPol activity, slowing fork progression and suppressing the formation of single-strand DNA gaps [97]. While PrimPol has received considerable attention, recent work from the Kannouche lab argues that there are other repriming activities. For example, they found that RAD51 was implicated in repriming, independent of its role in homologous recombinational gap filling and reconstruction of broken forks [85]. 

### 3.3. Translesion Synthesis

Another response to lesions similar to CPDs involves polymerases that perform translesion synthesis (TLS) [98,99] (Figure 1C). These have relaxed specificity for correct pairing and can insert bases across adducted bases [100]. While there are multiple TLS polymerases, for the purpose of illustration, we will focus on Pol η, encoded by the gene which is mutant in the variant form of Xeroderma pigmentosum (XPV). Reflecting the exposure to the sun throughout the development of life on earth, Pol η has evolved to correctly insert adenines at thymine CPD sites [101,102,103,104]. It is associated with the replisome under non-stressed conditions and further recruited following UV exposure [105]. Early work established the requirement for PCNA monoubiquitination to mediate the switch from the replicative polymerase to Pol η which would perform “on the fly” translesion synthesis [106,107]. More recently, reconstruction experiments with yeast replisome proteins support a more detailed model in which replisome encounters with a leading strand CPD inhibit Pol ε, but not the CMG which continues unwinding and generating single-strand DNA [108]. Pol δ then makes the first of two important contributions. It binds the stalled leading daughter strand, in effect protecting it from other polymerases, including Pol η. The formation of RPA-coated single-strand DNA signals monoubiquitination of PCNA by the E2/E3 ubiquitin ligase RAD6/RAD18. This drives the switch to Pol η which then synthesizes past the CPD, but dissociates rapidly from undamaged DNA, giving way to Pol δ again. Pol δ, which can synthesize faster than the CMG can unwind, extends the leading strand until it reaches the CMGE, which then resumes synthesis by Pol ε. 

Lesion skipping/repriming and translesion synthesis are competing options available to replisomes confronted with lesions that pass through the central channel of the CMG and then inhibit Pol ε [109]. The balance between the two depends on the genetics of the cell and the lesion. Electron microscopy of replication intermediates isolated from cells exposed to UV indicates that TLS by Pol η is the initial choice followed by repriming by pathways that involve PrimPol or other factors [85]. However, with bulky adducts, such as those of benzo[a]pyrene-diol-epoxide, repriming by PrimPol is favored. The gaps are then repaired by RAD5-mediated homologous recombination [110]. 

## 4. Encounter of the CMG with Putative Helicase Blocking Lesions

### 4.1. DNA:Protein Crosslinks

Except for the experiments with replicating plasmids with ICLs, the preceding discussions of repriming, translesion synthesis, and fork reversal have focused on experimental conditions absence of dNTPs, DNA adducts that uncouple leading strand synthesis from the continued activity of the CMG helicase. Lesions thought to block the helicase are also sources of replication stress and must be resolved to complete replication. In the ICL/plasmid replication experiments in Xenopus egg extracts cited above, fork convergence at the block triggers the eviction of both replisomes. This activates pathways that exploit fork reversal to remove the lesion and complete replication by repair synthesis [73,111]. However, replication of eukaryotic genomes features widely spaced origins as well as origin-poor regions. Thus, it is unlikely that all encounters with helicase-blocking lesions will be rescued by a converging fork. Furthermore, as noted above, fork reversal from a helicase-blocking lesion would require displacement of the replisome to allow access to reversal factors. Following removal of the block, the reconstructed fork would be without a replisome and unable to resume forward progress. 

Accordingly, the collision of a single fork with a helicase-blocking lesion necessitates other strategies. Fork barriers formed by proteins non-covalently bound to DNA, such as in ribosomal RNA gene loci, or experimentally established LacI-LacO complexes, might pause replisomes but can be disrupted by accessory helicases or, under certain experimental conditions, by the CMG [112,113]. 

On the other hand, covalent DNA: protein crosslinks (DPC) too large to pass through the CMG are more formidable barriers. These are formed by chemical agents such as aldehydes, chemotherapy drugs, abasic sites, ionizing radiation, and UV light [66,114]. At first glance, replisome stalling, then repair, and then restart of replication would seem to be an obvious strategy. Indeed, it was proposed in earlier work that the restart of synthesis would be preceded by either complete removal by a combination of proteolysis and excision repair [115], or proteolytic digestion yielding a DNA: peptide adduct sufficiently small to pass through the CMG [116]. However, recent studies demonstrate that the CMG can negotiate a DPC on either the lagging or leading template strand under conditions in which there is no proteolysis of the DPC.

Multiple groups have shown that intact DPCs on the lagging template strand do not block the CMG [48,117,118,119]. In light of the structural work showing the entry of duplex DNA into the N terminal tier of the MCM complex prior to strand separation, there appear to be at least two explanations for the bypass of the DPC (see [120]). One is that junctions between MCM subunits open transiently to permit the exit of the adducted lagging template strand out of the N-tier. This could be at the MCM3/5 gate [48,120], or between any subunits, the result of the breathing described by Langston et al. [53]. 

The second possibility is that the double-strand DNA with the DPC is unwound before entry such that only the leading template strand enters the CMG. This scenario was proposed several years ago by the Walter group who showed that streptavidin: biotin adducts on the lagging template strand were readily bypassed [117]. More recently, Kose et al. demonstrated that the kinetics of unwinding DNA with a covalent DPC on the lagging template strand were the same as with undamaged DNA. Since they expected that additional time would be required to evict a lagging template strand with a bulky adduct out of the N-tier of the CMG, they argued that the adducted double-strand DNA would be unwound before entry and that ring opening would be unnecessary [118]. As discussed below, duplex unwinding before entry into the CMG would alter the path of the lagging template strand, which eventually must be engaged by the Pol α- primase. 

CMG encounters with DPCs on the leading template strand have also been examined by the Walter group. They made the unexpected finding that the replisome could move past a DPC formed by a covalently attached M.HpaII methyl transferase on the leading template strand, ostensibly too large to pass through the CMG [121,122,123] (Figure 1, Top). Bypass was independent of the degradation of the DPC. Indeed, even in conditions in which proteolysis was possible, the DPC passed through CMG and then was digested by the proteasome and SPRTN proteases. Efficient bypass of the DPC was dependent on the RTEL1 accessory helicase. The construction of a plasmid with a bubble ahead of the DPC replaced the requirement for RTEL1, indicating that a single-strand bubble ahead of the replisome was important for the bypass. They proposed that RTEL1, operating on the lagging template strand, unwound DNA ahead of the DPC. Once clear of the CMG, inhibition of Pol ε by the DPC activated degradation of the protein portion by both SPRTN and the proteasome. This left a peptide adduct that was a substrate for translesion bypass synthesis by the TLS polymerases DNA Pol ζ and Rev1. 

Three notable observations were reported in this study. The first is the demonstration that the bulky DPC could be pulled through what was considered a too-narrow channel. The authors suggested that a gate in the MCM ring opened to permit passage of the DPC. This possibility has been discussed recently with further speculation that the movement of bulky blocks through the CMG might require the dissociation of Pol ε from the MCM complex [47]. Transient adjustments to the composition of the replisome might be necessary to support a structural flexibility that enables passage of what would seem an impassible block. This could reflect the opening of a specific gate or the transient gaps induced by “breathing” between MCM subunits discussed above [53]. Note that the adjustment would simply expand the diameter of the channel, there would be no movement of the strand out of the channel. Furthermore, while the premise of the experimental design was that the DPC was a helicase-blocking lesion, proteolysis and subsequent bypass of the proteolysis remnant were triggered by inhibition of polymerase ε, not by stalling the helicase.

The second is that bypass of the DPC required the RTEL1 helicase. Presumably, the DPC would be sensed by TIMELESS: TIPIN at the leading edge of the replisome. The proposal that RTEL1 would create a bubble on the “other” side of the DPC implies that RTEL1 would be loaded on the side of the DPC distal to the encounter of the adduct with TIMELESS: TIPIN. This suggests that the entry of double-strand DNA with the DPC into the groove on TIMELESS: TIPIN would trigger the loading of RTEL1. However, that raises another issue.

RTEL1 is a member of the XPD family of 5′-3′ iron-sulfur cluster helicases [124]. Typical biochemical assays of these helicases feature oligonucleotide substrates that contain a single-strand loading region and a duplex region that is unwound by the enzyme. The single-strand binding region is necessary because these helicases cannot unwind duplex DNA unless provided entry via a single-strand element. In the experiments with the DPC, it is not clear where and how this single-strand region would be generated. In contrast to bulky helix distorting/destabilizing adducts, the HpaII methyltransferase interacts with both strands of duplex DNA and increases duplex stability [118,122]. Thus, it seems likely that additional factors would be required to load RTEL1. One candidate would be MCM10 since it is located on the N terminal face of the CMG and interacts with RTEL1 in driving fork progression through replication barriers [45,120,125,126]. 

Finally, in models depicting the unwinding of double-strand DNA outside the CMG [117,118,122], the lagging template strand would not enter the CMG and thus not be evicted near the Pol α- primase. Presumably, this would describe the situation once RTEL1 forms the bubble. However, lagging strand synthesis does occur, so the lagging template strand and the primase polymerase must find each other. If these models are correct, this would suggest an alternative path to the polymerase for the lagging template strand. Alternatively, lagging daughter strand synthesis would resume when the lagging template strand would reenter the N-tier such that incoming duplex DNA would again be drawn into the N-tier and the lagging template strand evicted towards the Pol α- primase as before the encounter with the block [120]. 

### 4.2. DNA Interstrand Crosslinks 

A DNA ICL would appear to be the ultimate impassible adduct. For several decades after their identification, this intuitive view underscored their routine description as absolute blocks to replication [127,128]. However, unexpectedly, with a DNA fiber assay, we observed a restart of DNA synthesis past genomic psoralen ICLs [129]. Remarkably, this replication “traverse” of the ICLs was more than twice as frequent as the stalling of single forks or double fork convergence at ICLs. The restart was partially dependent on the FANCM DNA translocase and in part by the DONSON scaffolding protein [129,130]. The observation that FANCM and DONSON contributed independently to the restart revealed the contribution of two distinguishable versions of replisomes that encountered the ICLs. One contained DONSON and was largely active in euchromatin and the early S phase. The other was biased towards heterochromatin and the late S phase and was the recruitment target of FANCM following contact with the ICLs [130]. Subsequently, it was shown that PrimPol is involved in the restart of replication of replisomes that recruit FANCM following an encounter with ICLs [131]. 

The relationship between restart and ICL repair is also of interest. Traverse occurs in a few minutes and is independent of ERCC1/XPF and NEIL3, two activities known to be involved in the repair of psoralen ICLs [129,132,133,134]. However, following Okazaki fragment ligation, the traverse of ICLs would generate the same “X” structure formed in the experiments in Xenopus extracts [72]. The X structure is considered a requirement for subsequent ICL repair, in which reversal of one side of the X would establish a substrate appropriate for incision by XPF/ERCC1 [73]. Thus, restart of replication past the ICL would generate a structure appropriate for repair, without sacrificing continued replication. On the other hand, if there was no restart past a stalled fork, then completion of replication on the distal side of the ICL would likely require rescue by a downstream, possibly dormant, origin. 

## 5. Conclusions

In the preceding, we have emphasized events that follow the encounter of the replisome with covalent leading strand lesions. Lesion skipping/repriming, on-the-fly translesion synthesis, and fork reversal are all well-characterized pathways that promote the completion of replication. They are triggered by blocks to the leading strand polymerase, even by adducts presumed too large to transit the channels of the CMG. It appears that the evolution of the replicative apparatus favored the flexibility of the CMG over a stalled helicase [135] (Figure 1). A deeper understanding of the factors that underlie the response to replication stress should enable successful strategies to ameliorate the decline in replicative resilience associated with aging [136,137]. On the other hand, these insights would also be helpful in designing approaches to defeat tumor cells that overcome the extensive replication stress that characterizes unregulated replication.

## Figures and Tables

**Figure 1 cells-12-02607-f001:**
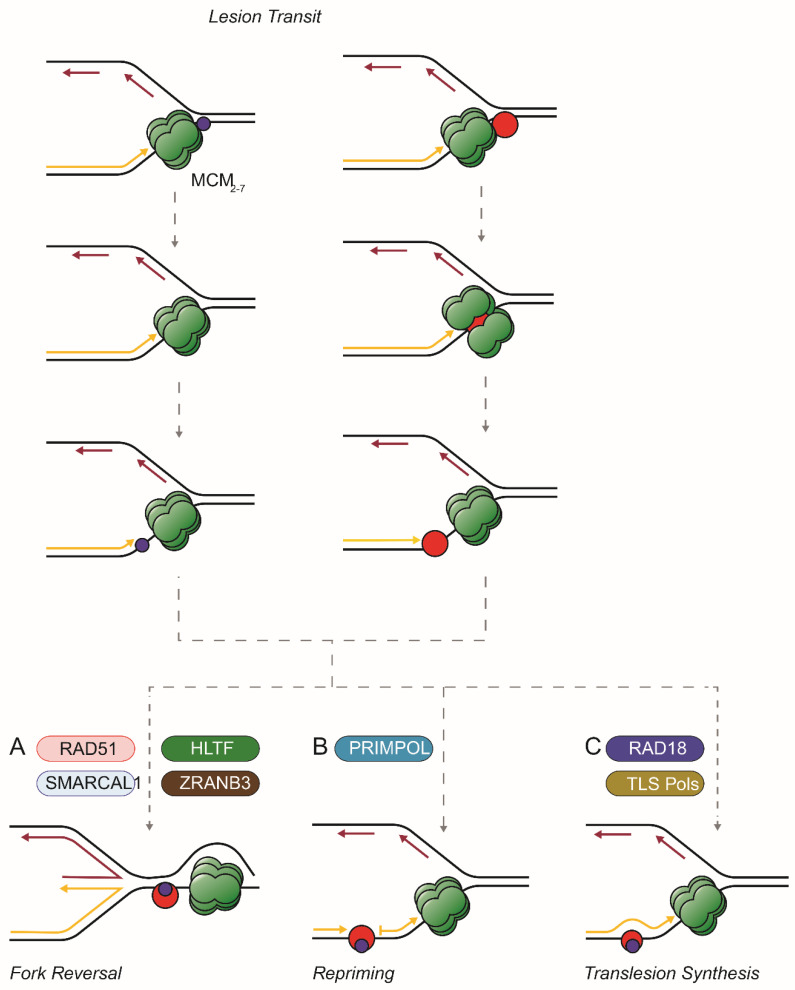
Options after the transit of large and small lesions through the CMG. Small (blue) or large (red) covalent adducts (such as a DPC) on the leading template strand are drawn through the replisome (here simplified to show only the MCM complex, in green). Then (adapted from [67]): (**A**) the fork might reverse driven by RAD51 and translocases such as SMARCAL1, ZRANB3, and/or HLTF. The replisome stays ahead of the lesion; (**B**) the replisome continues to unwind the parental duplex and repriming by PrimPol allows synthesis to continue leaving the adduct in a single strand gap; (**C**) RAD18 and translesion polymerases extend the daughter strand past a small adduct or, if a large DPC, reduced in size by proteolysis.

## Data Availability

Not applicable.

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
