# Peer review of "The Response of the Replication Apparatus to Leading Template Strand Blocks"

_cells, 2023, doi:10.3390/cells12222607_

Round 1
Reviewer 1 Report
Comments and Suggestions for Authors
Review of "The Response of the Replication Apparatus to Leading Template Strand Blocks" by Seidman and colleagues.
This is an outstanding review and discussion of the current state of the filed regarding how the MCM2-7 helicase is able to overcome various blocks to unwinding. The authors have expertly walked the reader through all of the caveats to helicase progression and the influential results that examined the mechanism of bypass or restart.
I only have one comment.
There should be more frequent referencing to the Figure 1 A, B, or C panels throughout. The text is very clear, but I kept wanting to have a visual representation to go along with the text. Currently, Figure 1 is only referenced in the final paragraph.
Author Response
We thank the reviewer for the helpful suggestion. We have included references to Figure 1 sections throughout the text as recommended by the Reviewer.
Reviewer 2 Report
Comments and Suggestions for Authors
I can only applaud the authors of this review, as it is – by far – the best and most insightful review on replication blocks that I have read in a very long time. Thank you for this exceptional read!
I have a few suggestions for editorial changes that the authors may or may not want to incorporate. All of these are minor and aimed at making the topic fully accessible to novices in the field.
Line 167: This is the only thing that needs to be fixed. There is an unformatted reference #348 (I believe it is reference 56.
Line 209: the authors talk about PrimPol. They may want to add that PrimPol activity is highly regulated and normally suppressed by a BRCA2-MCM10 complex. Kang et al. 2021 [PMID 34645815]
Line 219: across from should read “across” delete the “from”
Line 233: should read “by the E2/E3 ubiquitin ligase complex RAD6/RAD18” (just for accuracy)
Line 290: should read “for restart of DNA synthesis”
Line 291: give a brief definition of “paranemic”; many students might appreciate that.
Line 307ff: Thank you for pointing this out. There is also often a lack in correlating fork reversal or the inability thereof with a biological consequence (such as a change in viability/survival).
Line 325: “to resume forward progress” maybe change to “and unable to resume forward progress”.
Line 461: delete one period
Author Response
We thank the reviewer for their helpful comments that improve the manuscript. We have made all the changes requested - corrected typographical errors, and added additional text and citation regarding PrimPol as thoughtfully suggested by the Reviewer.
Line 167: This is the only thing that needs to be fixed. There is an unformatted reference #348 (I believe it is reference 56.
The citation has been corrected.
Line 209: the authors talk about PrimPol. They may want to add that PrimPol activity is highly regulated and normally suppressed by a BRCA2-MCM10 complex. Kang et al. 2021 [PMID 34645815]
We have added text and reference per the reviewer’s thoughtful suggestion.
Line 219: across from should read “across” delete the “from”
Corrected
Line 233: should read “by the E2/E3 ubiquitin ligase complex RAD6/RAD18” (just for accuracy)
Adjusted
Line 290: should read “for restart of DNA synthesis”
Modified.
Line 291: give a brief definition of “paranemic”; many students might appreciate that.
A definition has been included.
Line 325: “to resume forward progress” maybe change to “and unable to resume forward progress”.
Adjusted.
Line 461: delete one period
Corrected